# Pipeline In-Line Inspection Method, Instrumentation and Data Management

**DOI:** 10.3390/s21113862

**Published:** 2021-06-03

**Authors:** Qiuping Ma, Guiyun Tian, Yanli Zeng, Rui Li, Huadong Song, Zhen Wang, Bin Gao, Kun Zeng

**Affiliations:** 1School of Automation, University of Electronic Science and Technology of China, Chengdu 611731, China; mqp@std.uestc.edu.cn (Q.M.); bin_gao@uestc.edu.cn (B.G.); kunzeng@uestc.edu.cn (K.Z.); 2School of Engineering, Newcastle University, Newcastle upon Tyne NE1 7RU, UK; 3Shenyang Academy of Instrumentation Science, Shenyang 110043, China; zengyanli@hb-sais.com (Y.Z.); huadong_song800713@126.com (H.S.); 4PipeChina Northern Company, Langfang 065000, China; lirui@pipechina.com.cn; 5School of Automation, Nanjing University of Aeronautics and Astronautics, Nanjing 211106, China; zhen_wang@nuaa.edu.cn

**Keywords:** pipeline inspection, robot, pipeline integrity management, data management, non-destructive testing (NDT)

## Abstract

Pipelines play an important role in the national/international transportation of natural gas, petroleum products, and other energy resources. Pipelines are set up in different environments and consequently suffer various damage challenges, such as environmental electrochemical reaction, welding defects, and external force damage, etc. Defects like metal loss, pitting, and cracks destroy the pipeline’s integrity and cause serious safety issues. This should be prevented before it occurs to ensure the safe operation of the pipeline. In recent years, different non-destructive testing (NDT) methods have been developed for in-line pipeline inspection. These are magnetic flux leakage (MFL) testing, ultrasonic testing (UT), electromagnetic acoustic technology (EMAT), eddy current testing (EC). Single modality or different kinds of integrated NDT system named Pipeline Inspection Gauge (PIG) or un-piggable robotic inspection systems have been developed. Moreover, data management in conjunction with historic data for condition-based pipeline maintenance becomes important as well. In this study, various inspection methods in association with non-destructive testing are investigated. The state of the art of PIGs, un-piggable robots, as well as instrumental applications, are systematically compared. Furthermore, data models and management are utilized for defect quantification, classification, failure prediction and maintenance. Finally, the challenges, problems, and development trends of pipeline inspection as well as data management are derived and discussed.

## 1. Introduction

As energy demand increases, the energy production infrastructure expands correspondingly. Pipelines are predominantly used to transport oil, natural gas, water, and other important resources over long distances or between countries. They are recognized as one of the safest ways of energy transportation [1,2]. However, hazards such as metal loss, pitting and cracks might occur in a pipeline. These could result in personal injury or death, economic losses, and environmental damage [3]. Therefore, growing attention has been given in the research field to pipeline inspection and monitoring for condition-based maintenance and structural integrity management.

Pipelines employed in the natural gas industry are usually metallic, and the defects formation process consists of three essential components [4,5]. The first one is the inherent defects produced during prefabrication [6,7,8]. In this process, the steel pipelines are made of billet solidified by molten metal, and the billet will contain defects. Most of these defects will be removed when the head and tail of the billet are cut off, whereas there still exist a certain number of defects remaining in the billet. That is the inherent defects, including shrinkage cavities, casting hot cracks, air holes, inclusions, etc. [9]. Afterwards, these inherent defects in the billet will produce special defects in the rolling process of steel pipelines, including cracks, delamination, hairline, and so on. Then, the heat treatment, machining, coating and finishing process of rolled steel pipelines will produce discontinuities on the surface of the steel pipelines as well. This results in heat treatment cracks and coating cracks [10]. Most of these defects are distributed on the surface of steel pipelines as illustrated in Table 1. The second one is welding defects which occur in the process of pipe welding [11,12]. During the pipeline service, cracks and corrosion will occur at the junction of the pipeline matrix and weld, which will lead to a serious leakage [13]. These defects are the focus of non-destructive testing (NDT) and structural health monitoring (SHM). The third one is corrosion formed during service. There are two kinds of crack defect, stress corrosion cracking (SCC) and hydrogen-induced cracking (HIC) [14,15]. SCC is caused by the combined action of corrosive [16] environment and continuous tensile stress and it has been considered as one of the main failure modes in a humid environment. The microstructure, chemical composition, residual stress, applied load, grain boundary characteristics, the pH value of soil and transportation medium, and other parameters of different steel pipe materials affect the generation and propagation of SCC cracks [17]. HIC is a kind of stepped cracks that occurs when pipelines are exposed to hydrogen-containing medium and hydrogen precipitates into the steel during electrochemical corrosion [18]. The growth and development of these cracks eventually lead to the damage of pipeline steel. Corrosion defects can occur on both the internal and external surfaces of the pipeline. When the corrosive liquid is transported in pipelines, the fine sand structure and acid-base properties of transported materials will cause internal corrosion. Depending on the quality of the slurry and the speed of transport, corrosion may occur in different ways (uniform, corrosion) [4]. Moreover, due to the inherent defects, coating, or cathodic protection in the manufacturing process of steel pipes, corrosion can also be caused. Temperature, soil chemical composition, and the activity of microorganisms (such as bacteria or fungi) can be considered as the causes of external corrosion in buried pipelines. Among these, the percentage of corrosion is considered to be affected by microorganisms of 20% to 30%. Another type of corrosion is caused by high current, which is known as stray current corrosions [16,19]. These corrosion defects can take place on both the inside and outside of the pipelines and include pitting corrosion, exfoliation corrosion, intergranular corrosion, and crevice corrosion. Defects of pipelines produced during this operation are as shown in Table 2. When these cracks and corrosion defects extend and develop to a serious extent, then pipeline leakage will occur. Thus, to minimize these threats, detection and monitoring of pipeline integrity before failure including an understanding of defect progression, condition-based maintenance, and lifecycle management is important.

Research on in-line pipeline inspection has been intensified over the years. The NDT method is a common scheme for pipeline discontinuity detection and safety evaluation [20]. It refers to the testing without damaging or affecting the performance of the tested object. Conventional non-destructive testing methods include radiographic testing, penetrant testing, ultrasonic testing, visual testing, eddy current testing, and magnetic particle testing are recognized effective strategies [21]. However, different non-destructive inspection methods based on different principles have their characteristics [22] and usages for in-line inspection of pipelines. Moreover, different causes of defects will lead to different types of damage [4]. Therefore, the use of appropriate detection methods according to the specific detection requirements needs to be studied [12]. Due to the limitation of the detection principle and pipeline characteristics (such as size, medium property, etc.), the detection equipment and functions need to be designed according to the actual detection object as well [23]. In addition, current pipeline integrity evaluation and health management methods employ these historical data to identify and evaluate high-risk pipeline [24], while condition-based maintenance can be carried out. In overall, data management methods for defect quantification, identification, prediction and maintenance are crucial parts of pipeline integrity management. Thus, this paper systematically reviews in-line inspection of metal pipelines in association with robot-based instrumentation and data management, which are lacking in current literature. Specifically, it includes not only the state-of-the-art of research, but industrial applications. Work on detection and location of non-metallic pipelines can be found in reference [25].

Section 2 discusses the different NDT technologies for inline inspection. Section 3 introduces the device named the Pipeline Inspection Gauge (PIG) and a robotic inspection system. In Section 4, the data analysis methods and models are reviewed for defect quantification and classification. The challenges, problems, and development trends of pipeline inspection and data management are derived and discussed in Section 5. Finally, conclusions are outlined in Section 6.

## 2. Non-Destructive Testing (NDT) for In-Line Inspection

The scientific and engineering communities have presented various techniques to detect defects in an operational pipeline. These techniques can be grouped into three main categories for the maintenance of pipeline systems as shown in Figure 1 [29]. Technologies and defect detection and prevention include locating pipelines and underground facilities, determining excavation damage and encroachments to the right-of-way, leak detection, and damage mitigation. The pipeline and hazardous materials safety administration (PHMSA) of the U.S. Department of Transportation has recorded, excavation damage poses a leading threat to failure incidents of gas pipelines. However, leakage detection is usually an indicator of excavation damage and other threats. Thus, in order to reduce this damage and protect pipeline Right-of-Way (ROW), it is necessary to develop a locating equipment especially electromagnetic locators to identify the position of metallic pipelines. For instance, acoustic or ground-penetrating radar technologies are recent advancements. Among them, the damage mitigation and prevention technologies focus on minimizing the time and workload of taking measures to avoid the expansion of accidents. However, these technologies are hard to locate and protect inside the pipeline. Technologies for threats and integrity management contains inspecting existing defects (such as cracks, metal loss, rust and dents), making use of several devices in external corrosion direct assessment, internal inspection, pipeline detecting and monitoring, as well as stress analysis [30]. These technologies are the key points of threat identification and integrity management. Accordingly, plenty of in-line inspection (ILI) methods based on NDT have been developed to detect and quantify these defects or stress, and ILI is internationally considered as the most effective approach to detect and locate pipeline defects. As for risk assessment and information management, it covers various technologies for data visualization, asset tracking and traceability, geographical information system, risk assessment, awareness of response, as well as network and physical security. These technologies contain wide aspects of identifying high-risk pipelines for repair or replacement. Moreover, risk analysis using historical data depends heavily on the quantity and quality of the available data. Therefore, different NDT methods for ILI tools and data management-related defects quantification and health management are systematically reviewed and discussed in the paper.

The inspection strategies are as follows:
Magnetic Flux Leakage Inspection (MFL)

While the strong magnetic field acts on a ferromagnetic object, the geometrical discontinuity in this test specimen will result in the flux leakage from the tested piece into the air [31]. The magnetic flux leakage (MFL) is monitored by magnetic sensors around the circumference to estimate the dimensions and types of the defects. Although the phenomenon of magnetic flux leakage is easy to understand, the design and analysis of the MFL system involve the complex interaction of excitation, flux leakage, and material with defects. As a classical NDT method, much work has been done already. Wilson et al. [32,33] studied the visualization imaging and description of defect damage using 3D MFL imaging technology. Pham et al. [34] proposed a planar Hall magnetoresistance sensor and applied it to oil and gas pipeline inspection based on MFL. The prototype sensor has potential applications to detect shallow defects in the near-surface, subsurface, or external surface, corrosion in particular. Due to traditional MFL ILI being employed an axial excitation strategy that cannot identify the narrow cracks along this direction, an axial crack detection model was established in [35] through a linear magnetic dipole model with the circumferential excitation method. Moreover, Yang et al. [36] summarized the principle and influencing factors of internal detection of MFL technology for oil and natural gas pipelines. They elaborated the research status of key technologies such as axial excitation and circumferential excitation in MFL internal detection of pipelines and compared the detection ability of magnetic flux leakage internal detectors. Azad et al. [37] designed and optimized an MFL coil sensor for cross-sectional metal loss detection. The optimal parameters are obtained from numerical study. Azizzadeh et al. [38] used 3D finite element analysis to acquire three axial MFL components for quantitative analysis pitting sizes. Except for defects recovery based on MFL signal, Mukherjee et al. [39] provided an adaptive channel equalization algorithm to remove the noise caused by sensor movements and manufacturing imperfections.
Ultrasonic Inspection (UT)

Ultrasonic inspection (UT) is one of the principal directions of conventional NDT methods to identify the defects of high-frequency sound waves on materials or their surface. When ultrasonic waves move through an object, they consume energy and reflect off the surfaces. Analysis of the reflected sound can determine the presence and locations of the discontinuity or defects. UT can easily determine cracks, crevices, metal losses, and other discontinuities at different depths inside sample since ultrasonic sound has characteristics of reflection, diffraction, and transmission [40]. However, conventional ultrasound requires a couplant to fill the gap between the probe and the surface of the workpiece to be tested. It is difficult to detect the workpiece with complex shapes and irregular shape, and it cannot effectively detect the pipe surface and surface fatigue cracks. Therefore, ultrasonic phased arrays, guided wave detection technology, electromagnetic ultrasonic testing technology, and laser ultrasonic testing technology are proposed to improve the inspection capability. A phased ultrasonic array (PAUT) [41,42] can overcome the limitations of traditional UT. In PUAT, the transducer is composed of several independent piezoelectric chips. According to certain rules and time sequences, the electronic system is used to control and excite each chip unit to adjust the position and direction of the control focus. PAUT [43,44] total focus imaging technology [45] based on full matrix data for virtual focusing has the characteristics of high accuracy and flexible algorithm. It is effective for the detection of complex shape defects. At present, the multi-probe ultrasonic testing system is gradually replaced by the ultrasonic phased array system. The new automatic ultrasonic testing system can complete a large number of tasks undertaken by conventional probes by a pair of phased array probes. However, PAUT needs a lot of data analysis for pipeline inspection, and this method also has limitations for high-speed pipeline inspection. With the rapid development of information technology, UT is gradually moving from traditional detection to automatic non-destructive quantitative evaluation and structural health life prediction [46]. Guided wave UT (GWUT) [47] is one of the latest methods for bridging NDT and SHM, and can be used for pipeline monitoring to achieve an extensive structure area inspection from a test point [48]. Cawley et al. [49] researched the guided wave propagation on the pipe wall and looks for the reflection of the defect [50,51].

The generation mechanism is similar to that of a Lamb wave on a thin plate. It is caused by repeated reflection in the medium with limited space as well as further complex superposition interference and geometric dispersion [52]. It has the characteristics of long-distance detection and no need to scan point by point on the detection surface [53]. When transferring this technology from laboratory to application field, challenges [48] such as pipeline filling, coating other structures with complex geometry were solved step by step. Moreover, the physical characteristics of bending mode and symmetrical mode have been understood and the corresponding guided wave detection system was built [54,55]. The defect detection and positioning of various waveguide structures were successfully carried out [56]. The mechanical properties of new materials were quantitatively characterized as well [57]. Furthermore, detection devices with the guided wave excitation method of pipeline were designed in [58]. Apart from this, many studies were carried out in [59,60,61] concerned with the mode conversion characteristics of guided waves passing through the bending parts of pipes. An electromagnetic acoustic transducer (EMAT) is based on electromagnetic coupling [62]. Under alternating magnetic fields, the eddy current is produced in a metal conductor. At the same time, the metal medium generates a stress wave due to the Lorentz force acting on the current of the non-parallel magnetic field. The stress wave whose frequency is within the range of ultrasonic wave is the ultrasonic wave. In addition, due to the reversibility of this effect, the return sound pressure makes the particle vibrate and changes the voltage at both ends of the eddy current coil under the action of the magnetic field. The detection signal is received and amplified by the receiving device to analyze the defect information [63]. Thus, this technique does not require any couplant medium and can produce a wide range of patterns [64,65,66,67,68]. It shows the potential in austenitic weld inspection [69,70,71]. It can be applied for non-contact inline pipeline inspection with low efficiency.
Eddy Current (EC) Technique

Since there exists a blind area of surface inspection in conventional ultrasonic technology, the EC method based on electromagnetic principle is proposed for pipeline surface and near surface defects detection [72]. It is widely used for crack detection since it is sensitive to conductivity, permeability and thickness variation of materials [31]. When an alternating magnetic field interacts with a conductive test workpiece, then the distribution and size of eddy current are changed. It is affected by the characteristics of materials, such as conductivity, permeability, defects size and shape. Through measuring the magnetic field changes caused by an eddy current received from detector coils, the conductivity and defect characteristics are obtained [73]. EC is a non-contact measurement method, which can be used for high-speed inspection [74]. However, it can only inspect the surface or near surface structure of conductive materials due to the skin effect. In addition, eddy current testing is greatly affected by lift-off distance, that is, the distance between the probe and the surface of the pipeline to be tested [75]. Besides, remote field eddy current (RFEC), eddy current array (ECA) and pulsed eddy current (PEC) are other research hotspots developed on the basis of conventional eddy currents. RFEC is a kind of low-frequency eddy current that can penetrate the tube wall twice [76]. The detector coil is usually placed about twice the diameter of the pipeline from the excitor coil. With a low-frequency alternating current, the magnetic field signal passes through the pipeline twice and then returns to the detector coil for receiving [77]. Thus, it can effectively detect the internal and external wall defects or the thickness reduction of the metal tube [78]. In addition, traditional EC usually applies a single probe with mechanical scanning over a surface, which results in a complex scanning path and decreases the detection sensitivity as well as reliability [79]. ECA arranges multiple elements on the surface of the tested pipe through certain settings for inspection, which provides time-saving and a good solution for complex scanning paths [80]. The conventional eddy current is affected by skin effect, and the detection depth is limited by the excitation frequency. PEC uses the pulse excitation signal to induce the transient current in the tested object and it has rich spectrum content [81]. By analyzing the frequency variation of the transient flow, the defect detection, attribute characterization and condition evaluation of the specimens with different depths are realized [82]. In application, Denis et al. designed and optimized the PEC probe [83]. In their study, the probe’s sensitivity is improved at a higher lift-off with a certain coil gap. For higher sensitivity and imaging accuracy, Li et al. [84] proposed a pulse-modulation eddy current technique (PMEC). It can enhance the evaluation sensitivity to external corrosion and the accuracy of corrosion imaging compared with a traditional PEC. To provide solutions on probe scanning for long-distance inspection, Chen et al. [85] proposed and designed a magnetic force transmission ECA probe. Moreover, a lot of research has been done on the corrosion detection of pipelines with cladding and the distinction between internal and external wall defects. Sun et al. [86] proposed the application by RFEC in pipeline testing and made a relatively complete theoretical analysis. Kim et al. [87] presented a prototype pig based on RFEC. They employ a multi-phase rotating magnetic field in the remote region for SCC detection. Fukutomi et al. [78] used an electric scalar potential and a magnetic vector potential to analyze the electromagnetic field of RFEC. Afterwards, they optimized the RFEC probe and achieved micro defects on non-magnetic steam generator tubes detection.
Eddy Current Pulsed Thermography (ECPT)

Eddy current pulsed thermography (ECPT) is based on the phenomenon of eddy current and Joule heat in electromagnetism. It uses an infrared thermal imager to obtain the temperature field distribution and conduction of conductive specimen under the excitation of pulsed eddy current due to Joule heat. Defects detection can be achieved through analyzing and processing the multi thermal images [88,89]. Compared with other infrared thermal imaging methods [90], ECPT uses pulsed electromagnetic excitation for volume heating. It has the characteristics of electricity, magnetism, and heat, rich transient information, high spatial resolution, and high detection sensitivity for near-surface depth defects. The induction heating is concentrated on the defect, which increases the temperature contrast between the defect and the non-defect area. It can improve the signal-to-noise ratio and the detection sensitivity of small defects as well, multiple defects in complex geometric geometry in particular. Vrana et al. [91] developed a fixed and portable induction thermal imaging detection system, which can effectively detect cracks with a depth of 100 μm. Oswald et al. [92] carried out induction thermal imaging simulation and experimental analysis on metal surface defects. Moreover, they studied the temperature distribution of surface defects of materials with different electromagnetic parameters. Furthermore, they discussed the open defects of induction eddy current imaging on a steel plate and researched the influence of defects sizes and defects orientation. In addition, Tian et al. [93] developed the internationally leading eddy current pulse thermal imaging nondestructive testing and evaluation technology experimental platform. It is based on the fusion excitation of electromagnetic and thermal multi-physical field effects. It can be applied for weld inspection during installation and external pipeline inspection of maintenance.
Magnetic Barkhausen Noise (MBN)

Magnetic Barkhausen noise (MBN) detection technology has been widely used in the evaluation of (residual) stress and fatigue aging of ferromagnetic materials. The principle of this method is to study the signal characteristics of magnetic or acoustic emission. They are caused by domain reversal in the magnetization process and reflect the microstructure and stress distribution of ferromagnetic materials. Jiles et al. [94] studied the influence of residual stress, elastic and plastic deformation on the Babbitt effect in steel and the influence of microstructure of steel sample on MBN. According to the mechanism of MBN, it can not only detect the stress but also detect the microstructure of ferromagnetic materials such as fatigue life and small cracks. Jancarik et al. [95] analyzed the lift-off effect of building steel samples with different carbon content. It revealed that the slope of MBN amplitude distribution was not affected by the lift-off effect whereas the sensitivity became smaller. However, the repeatability, under different magnetization and influence of environmental condition and material variation, is not well understood [96]. It can be applied for stress measurement and inspection of material health states.
Radiography Testing (RT)

As one of the five conventional NDT methods, radiography testing (RT) is widely used in the industry. It provides an internal view of the inspected specimen and records permanent images. The significant advantage is that it does not require surface treatment or insulation removal [97]. Moreover, it is not sensitive to naturally growing external deposits on the pipeline surface [98]. Researchers around the world have been applying this method for welding inspection and developing techniques for supporting automatic radiographic [99,100]. Thus, automatic inspection from RT images with computer-aided design to overcome limitations of human interpretation is a hotpot in this field [101]. For example, Yazdani et al. [102] aims at the problems of unclear edge and low contrast of X-ray images and proposes an automatic image-processing technology for noise removal and image quality enhancement. Although RT can intuitively display images of defects, it still contains major disadvantages of the health risk associated with the radiation. In addition, it is difficult to recover the developing and fixing solution, and direct discharge will cause environmental pollution [3]. It can be applied for inspection during installation and maintenance of pipelines.
Acoustic Emission (AE) Inspection

Acoustic emissions (AE) is a kind of physical phenomenon that the transient stress wave produces by the rapid release of elastic energy when the object is subjected to deformation or external action [103]. In particular, it is a dynamic NDT during the internal structure of components or materials, defects, or potential defects in the process of motion change. Through receiving and analyzing acoustic emission signals of crack propagation, plastic deformation, or phase transformation, the material performance or structural integrity can be evaluated. Therefore, AE is a promising method in pipeline monitoring and researchers have done a lot of work. Quy et al. [104] collected AE signals and used the time difference of arrival to achieve emission source location in a pipeline conveying a liquid under high pressure. Then through calculating the returned sound source, irregular structural changes in the material such as cracks can be detected and located. Paton et al. [105] studied the acoustic characteristics of steam pipe materials of hot industrial steam by AE and analyzed the possibility of continuous monitoring and fracture load prediction of the pipeline. However, the AE signal is easily affected by the material, geometry, and sensor of the measured object, and it is difficult to directly build a connection between AE signals, fracture and material conditions. Therefore, a suitable signal processing method is required for clustering AE activity and identifying the features related to the fracture event [106,107].

NDT strategies like penetrating testing (PT), magnetic particle testing (MT), visual testing (VT) and are commonly used in the industry. These different methods have their own merits and limitations as described in Table 3. However, pipeline transportation lines are usually installed underground, and it is quite difficult and expensive to access from the outside. Therefore, the ILI devices are widely used for the inspection of the buried pipelines [108]. The instrumentation systems can be operated in oil and natural gas pipelines, water pipes, submarine systems, or any other pipeline systems for which specific inspection is required [109]. In this sense, the physical data about the pipeline integrity are collected and recorded using smart devices moving in the pipe [110]. However, due to the limitation of pipeline volume and length, not all methods are suitable for in-line pipeline inspection [111]. Up to now, only based on ultrasonic and electromagnetic ultrasonic, magnetic flux leakage, eddy current testing methods have successfully developed internal testing tools for in-line inspection of pipeline.

## 3. Pipeline Inspection Gauge (PIG) and Other Un-Piggable Robotic Inspection Systems

### 3.1. Pipeline Inspection Gauge (PIG)

In this section, we provide a detail review of the application on state-of-the-art ILI techniques and instrumental systems for pipeline inspection. According to the type of pipeline, these can be divided into piggable pipelines and un-piggable pipelines. Therefore, the detection tools involved are inevitably different. For example, various widely used Geometry Pig (GP), MFL Pig, UT Pig, EMAT PIG, EC Pig, Integrated Function Pig and Specific Function Pig are contained for piggable pipelines. The general ILI system fundamental structure and intelligent PIG types are illustrated in Figure 2 [110]. The other one is represented by a closed-circuit television camera (CCTV) and smart ball for un-piggable pipelines. Thus, these two kinds of detection technologies will describe with two subsections, respectively.

In recent years, significant progresses in the research and development of instruments and equipment have been made by many companies. For instance, T.D. Williamson company introduced a SpirALL Magnetic Flux Leakage (SMFL), which describes the advantages of pipeline detection based on the SMFL structure, and complements the shortcomings of the uniaxial magnetic field [114]. The Rosen company [115] provided an MFL and EC tool, which combined the MFL and EC methods to improve the measurement performance of thick-walled pipelines. These two technologies are well-established in the industry and present a robust inspection performance with high sensitivity and high precision. The approach not only ensures high sensitivity and high accuracy of EC in scanning abnormal metal loss of the inner tube, but also provides an amount of comprehensive geometric inspection information [116]. In addition, simultaneously employing MFL provides reliable additional information for mid-wall and exterior features [117].

Table 4 investigates the emerging commercial PIG technologies for anomaly detection and characterization. ROGEO Untouched geometry PIG (GP) is a tool that determines possible deformations developed by ROSEN company, such as dents caused by external force damage [110]. The main purpose of the geometry devices is to collect data relating to the physical shape or geometry of pipelines. The MagneScan SHR MFL detector developed by GE PII company is suitable for the pipe diameter range of 76–1422 mm [4]. The high field “Speed-stable” magnetizer enables the detection speed, which can reach 5 m/s, and 216 Hall effect sensors have been integrated for super high resolution to identify and size significant corrosion. LineExplorer UCM (a product configured for metal loss and crack inspection) [118] was developed by NDT-GLOBAL company, it has a special configuration that unites metal loss and cracks detection and available for pipeline size 20″ and above. In addition, Rosen, a leading industry organization, produced an EMAT PIG which is characterized by high reliability detection and accurate continuous measurement of all critical crack discontinuity, coating disbandment as a precursor of cracks can be detected reliably [119]. This PIG can apply for pipe diameter covered 12″–48″. As for EC tools, Rosen is dedicated to corrosion detection and heavy-walled pipeline inspection with eddy current testing. It has developed a pipeline eddy current internal detector for metal loss, which is a combination of a deflection sensor that allows simultaneous measurement of the inner pipeline profile. Therefore, not only corrosion defects but also deformations are captured in a single run. I2I Pipelines company [120] introduced a mandrel-style smart pig that can be run in the same way as a conventional cleaning pig. This pig has electromagnetic sensors embedded into the polyurethane. The array of electromagnetic sensors detects shallow internal corrosion and fatigue cracking (SICC) in dry gas or multiphase pipelines. For high-resolution imaging of internal corrosion and providing internal initiated and relative stresses. Meandering Winding Magnetometer Array (MWM-Array) technology is used by JENTEK in an ILI tool [121]. This tool has extremely high data rates capable of high-resolution imaging at a speed of up to 10 m/s and the tolerances for variable travel speeds variable sensor to pipe wall gaps. Considering the advantages and limitations of different single NDT and E techniques, better results can be obtained if they are properly combined. One example of the advanced combined NDT device is explored by the T.D.W company. It integrated the pipeline deformation (DEF), SMFL, MFL, lower magnetic field (LFM), and EMAT. The paper [122] introduced the advantages of this multiple NDT technology in detecting the pipeline defects in particular.

The integrative approach combined MFL and EC is recent research progression of authors [123,124]. The Shenyang Academy of Instrumentation Science, China, developed the magnetic eddy current (MEC) ILI tool. This tool equipped with special designed MEC probes as shown in Figure 3 and Table 4. It realizes the inspection of compound pipelines with stainless steel and carbon steel in two layers. Another emerging commercial PIG technology is the specific function device. Apart from this, a cathode protection current measurement ILI system was developed by Baker Hughes, which can capture data that verifies the effectiveness of cathode protection [125]. However, different application scenarios and inspection objects were considered when adopts different NDT technologies with these emerging commercial PIGs. In practice, more factors and types of defect to be inspected need to be taken into consideration. Therefore, Table 5 details the three type defects and provide brief advice on ILI PIG selection.

### 3.2. Other In-Line-Inspection Systems Suitable for Un-Piggable Pipelines

ILI tools have been used for around 50 years and have proven reliable and accurate. Unfortunately, not all types of pipelines can be designed to be piggable. If such a pipeline has not been designed with launchers and receivers, there will be limited access to PIG. Also, some pipelines are un-piggable due to wear or damage affecting their pigging capacity. More so, bends, external damage to the pipe, the build-up of solids on the pipe bore, and changes in pipeline cross-section can make pipes un-piggable [128,129]. In fact, nearly half of the world’s petroleum or natural gas pipelines have until recently been recognized as “un-piggable”. Below we review some in-line inspection systems suitable for inspection of Un-piggable pipelines.

The Smart-Ball tool from Pure Technologies, Ltd., is one of the most recently developed ILI technologies [130,131]. Figure 4 is a Smart-Ball produced by Pure Technologies [130]. The device is composed of instrumented an aluminum core in a polyurethane shell and a series of instruments containing an acoustic data acquisition system. An acoustic data acquisition system can monitor the leakage when the ball passes through the pipeline. Since it is not a full diameter instrument and the diameter of the ball is smaller than that of the tube, this smart ball is different from the traditional inspection pig that uses the cup to seal the pipeline. It rolls silently in the pipe, without mechanical noise, and has unparalleled acoustic sensitivity. Pinhole size leaks can be detected as low as gallons per minute (0.1 L/min) and the leakage positioning accuracy is within 3M. The sensitivity is exponentially higher than (Computational Pipeline Monitoring) CPM-based leak detection systems. What’s more, the advantage increases along with pipeline diameter. In addition, this device can be deployed and recovered using existing pigging facilities. In non-pigging pipelines, ready-made accessories can be used to start the equipment. Therefore, Smart-Ball technology could be complemented for ILI systems in un-piggable pipelines, oil and water pipelines in particular.

In-pipe robots have a long history of development [132] and, according to movement patterns, can be classified into wheel type [133,134,135,136], track type, walking type [137], screw-type [138,139], and inchworm type [140,141]. In-pipe robots compared with ground robots, the most significant difference is that its task space is the pipe, which needs a closed space force. Oil and gas pipelines distribute at least hundreds of kilometers in a three-dimensional space, there are vertical and level, elbow, tee. Because of energy limitations, existing in-pipe robots are all linked with an electrical cable, which is not suitable for a long-distance and in-service pipe. In fact, In-pipe robots only operated in a several kilometers pipeline, which often is used to inspect some un-piggable short pipeline. In-pipe PIG systems are mostly made up of the mechanical system and inspection system, where the mechanical system supply force to move forward and inspection system scan the pipe flaw. The inspection system can be equipped with different techniques like visual, laser, sonar, and other NDT techniques. In [142,143], authors report an un-piggable case where a special robotic unit based on a multi-trotter crawler (MTC) was combined with a bidirectional MFL inspection module in a 2 km 10” pipe.

However, CCTV is still the most common in-pipe robotic method, which is made up of the following hardware: illumination and lighting system, imaging sensors and cameras, digital camera interfaces, and computation units. Figure 5 is a typical CCTV [144]. The light produced by a light source is reflected from the detected object. The selection of the light source relies on the types of detection tasks, the scale of the inspection region and requirements of wavelength and brightness [20,145]. A camera or imaging sensor then captures the reflected light for further analysis. There are two main types of image sensors, i.e., charge-coupled apparatus and complementary metal-oxide-semiconductor. The camera interface is one of the most critical choices when building up a vision system. Common interfaces contain capture boards (frame grabber), USB, FireWire, GigE, Camera Link, and CoaXPress. A computerized optical microscope is used to obtain the images of the surface and the same images are fed into MATLAB software for further investigation. In emerging research, M.S. Safizadeh [20] proposed an intensity-based optical system for internal pipe inspection. Near-infrared reflectography and infrared thermography were also used for NDT and E of artworks in [146,147]. In [148,149], a laser profiler was used to improve the quality of the pipe images obtained by CCTV technology. In order to measure surface quality accurately, a computer vision system was used in [150,151,152,153] to characterize the nature of a surface texture. Nevertheless, an appropriate illumination system, robust image processing algorithm are still the challenges of optical and vision inspection system [154]. Its main limitation lies in the inability to detect and evaluate subsurface defects.

## 4. Data Management

After collecting the relevant data through detection and identification devices, data processing needs to be implemented to minimize data errors and extract helpful information. However, there is position displacement between the instrumental system and the pipeline in the actual detection process. Collected signals are inevitably influenced by various factors, such as electrical conductivity, magnetic permeability, lift-off distance, thickness and inhomogeneity of the pipeline, or other noise. How to obtain useful information and identification from the signals in the noise and the low-level signals under certain conditions is the basis of pipeline integrity analysis. The incorrect identification and classification of defects information will lead to serious inaccuracy in predicting defect growth. Consequently, the correct signal processing, signal analysis, feature extraction, and classification models are necessary to obtain the desired parameters. In addition, corrosion and crack shape from signals can better support defect growth prediction and integrity management. This section reviews the methods of defect feature extraction, quantitative classification. Then the means of pipeline defect growth prediction, artificial intelligence (AI) for decision making, and condition-based maintenance through multiple data are considered as well. Finally, the construction requirements of an integrated data management system or cloud network are analyzed.

### 4.1. Defect Quantification and Classification

Defect qualification and classification is a form of inverse process in electromagnetic nondestructive testing and it is a crucial element to determine the maintenance operations [155]. Electromagnetic inverse problems can be divided into two categories. Firstly, they belong to parameter identification and second to optimization design. The problem of parameter identification is to inverse or reconstruct the source parameters or the electromagnetic properties of the medium under the given experimental results and parameters. The position identification, shape identification, and medium parameters (conductivity, permeability, dielectric constant, etc.) are included. The optimal design is to provide the expected performance index of an electromagnetic system and then to achieve this goal by optimizing the parameters. Thus, the state or variation of defects and pipeline attributes (geometric attributes, physical attributes) can be determined through signal processing methods or by optimizing the performance of the detection sensor according to the received data.

A classifier will provide reliable information on the location and type or dimension of a tested defect. This machine learning process would be advantageous to reduce dependency on human interpretation skills. Up to now, two major groups of classifier include supervised and non-supervised methods [156]. Training data is labeled based on the class, such as defect types or locations in supervised classification. On the other hand, the training data do not need to be labeled in unsupervised classification. Support vector machine (SVM) [157], linear discriminant analysis (LDA) [158], K-nearest neighbor (KNN), K-means [159] and Bayes [160] are widely used methods for classification. For instance, SVM techniques in conjunction with Fisher linear discriminant analysis (FLDA) for localization and classification of defects were employed in [161]. Zajam et al. [162] studied a supervised machine learning method and wavelet analysis for various types of defects detection in natural gas pipelines. Khoa Le et al. [163] provided an online learning mechanism called SVM-CBR (case-based reasoning system). In this mechanism, the learning SVM model needs to be invoked at each stage. This integrated method has been successfully applied for pipeline defects inspection. Layouni et al. [164] used machine learning and pattern-adapted wavelets to inspect and size the metal loss in pipelines. Rostamabad et al. [165] developed a machine learning program with feature extraction, selection, recognition and regression for major pipeline defects based on MFL signal. A Bayesian framework is proposed in [166] to find the break point of signal intersection automatically in an acoustic emission signal source. Moreover, K-means, as an unsupervised method, has been applied for estimating the size of cracks in aircraft structures through Giant Magneto Resistance (GMR) sensing imaging data [159]. In addition, deep learning can solve highly complicated problems and it is one of the fastest developing fields in computer science [167]. Therefore, this method is effective in achieving automatic defect inspection and identification. Two deep learning approaches for thermographic image reconstruction were studied by Kovács et al. [168]. By comparing these with other methods, the hybrid deep learning approach has an outstanding performance. Ruan et al. [169] proposed a Defect-Detection Network (DefectNet) with a joint loss Generative Adversarial Networks (GAN) framework for infrared thermal images. Through modifying the GAN loss and penalty loss, the training process detection rate is significantly improved. Hu et al. [170] embedded a sequence-PCA (Principal Component Analysis) layer and designed a new attention block to a deep learning network for automated thermography defects detection. The tested results verify that the proposed model can capture semantic information better and improve the detection rate in an end-to-end procedure.

### 4.2. Pipeline Defect Growth Prediction and Condition-Based Maintenance

It is vital to have pipeline inspection data, which include strong defect information, for example, type, size, and relative position of defects. Information of the pipeline (manufacturing information, installation information, use information) where the defect lies is also included. Furthermore, the information of the pipeline network (climate, soil), etc., can be acquired, also. The above information would be accumulated with the pipeline in service. Therefore, the study of the relationship between these data and the establishment of a model to assess the health of the pipeline through data mining can predict the development of defects and pipeline leakage. To forecast the remaining useful life of pipelines, approaches are required to develop relating to the following three aspects [112]. First is the number of defects. [112] gives the understanding of the performance of ILI tools and the number of defects determined to the probability of detection (POD) of detection. By fitting the POD with different feature dimensions, the POD curve of the inspection tool can be obtained. Although different detection devices have different POD curves, the probability of detection cannot reach 100%. A few methods can be taken advantage of to update the actual number of defects, such as Bayesian and non-homogeneous processes. On the other hand, the types, sizes and locations of defects are important to be identified and quantified correctly. Finally, the correlation of defects cannot be neglected. For instance, investigation of crack interaction should be presented for better prediction. The failure models of different types of defect should be thoroughly studied. The remaining service life of the pipeline is determined based on degradation models and defect sizes, which is detected by inspection tools.

Thus, the gathered data should be organized, analyzed, and stored in an established database to successfully develop prediction tools. Correct analysis of available data and information of the tested system from both structural and operational perspectives is vital to determine the performance of the system and highlight any possible defects. Through feature extraction, feature selection, feature fusion and information fusion, these features are analyzed and fused for condition-based maintenance (CBM) [171]. That is to say, defect growth prediction and condition are obtained; predictive maintenance can be arranged accordingly.

The condition of the assets and decisions concerning the prioritization of inspection, repair, or renewal of pipelines can be determined through condition assessment models. Different models and techniques can be used to estimate pipeline failures caused by a various source of damage. Three types of available model, namely: physical, statistical, and artificial intelligence, are conventional assessment methods [172]. Physical models account for a distinct quantitative relationship between degradation factors and the condition of pipelines in service, whereas the uncertainty of the deterioration process is not considered. In contrast, the uncertainties through employing probability-based equations are considered in statistical models. Appropriate probability distributions rather than deterministic quantities represent the model variables in these models. Moreover, artificial intelligence models are recognized to be data-driven and not model-driven, where the mathematical relationships between the failure factor and condition data are evaluated by “learning” the failure behavior from collected data.

Physical models concern the physical mechanisms of the failure process of pipelines, whereas inherent uncertainty in the failure process is not accounted for. These models are based on the physical properties and mechanics of a certain phenomenon, which are known as deterministic models as well. Meanwhile, linear, non-linear fitting equations and single value degradation models related to the failure of the pipelines are available [173]. By assuming the linear process in defects growth, the linear growth rate models are applied to forecast the depth of defects in linear modelling. Wang et al. [174] proposed a general linear model framework and it can approximately describe a wide range of pipeline condition assessment and defect detection problems. The system response is determined by a linear function with pipeline properties at discrete locations along the pipe. Afterward, the pipeline characteristics are reconstructed by fitting the measured response with the least square method. As for estimating the external corrosion of buried and aged oil and gas pipeline, non-linear functions are introduced to describe the nonlinear relationship between soil factors and pit depth as the influence of the factors on the pit depth growth is complexity [175]. The final model is the weighted sum of these individual non-linear response functions representing its partial effect on the observed pit depth. On the other hand, the single value models are another deterministic approach extensively used, which assume a steady degradation growth rate over the analysis [176]. However, the independence of the age and depth of defects are limitations of this modelling.

Statistical models are composed of a set of a probability distribution, which are used to represent patterns of variability that random variables may display. In these cases, historic data are used e.g., the probability of uncertainties of pipeline occur is taken into consideration. Pesinis et al. [177] presented a statistical model which adopts a parametric hybrid empirical hazard model complemented with the non-linear quantile regression for reliability analysis and prediction. The model is segment-based and non-uniform Poisson processes/Poisson square wave processes are used to model the defect to estimate rupture probabilities due to external metal loss corrosion. A discrete Ferry–Borges stochastic process is used to model the internal pressure load. Then the reliability of gas pipelines is estimated based on historical failure data and the theory of structural reliability. Unlike the statistical method based on historical data, a time-dependent physics-based probabilistic approach was employed to determine the failure probability of pipelines [178]. The impact of corrosion, external loads, burst failure, etc., is considered as well. To derive different corrosion rate (CR) distributions, various corrosion growth models are involved and are not limited to probabilistic models in [179], such as a single-value distribution, linear growth corrosion model, time-dependent, time-independent model and Markov chain. Then a Monte Carlo reliability framework that combining these CR distributions was developed and applied to synthetic and field-collected corrosion data. The use of comprehensive data helped to evaluate the performance of each CR model and to consider corrosion defects of different sizes and ages. In addition, a gamma process-based corrosion growth model is explained in [180]. Due to the monotonic increasing nature of the gamma process, it is suitable for degradation mechanisms such as wear, fatigue, corrosion and creep. In addition, the interaction process between complex environmental conditions and pipelines can be studied by finite element simulation. Moreover, the uncertain deterioration factors such as the physical properties of soil, corrosion patterns, etc., can be considered. Defects and stress evaluation and predictive maintenance with both deterministic and probabilistic approaches linked with finite element simulation are investigated in [181].

The analysis of a large number of pipes is often intensive and time-consuming because the pipeline work is often large and complex. Data-driven methods are a research hotspot in the era of big data, which is based on data to load the appropriate algorithm and obtain the optimal model. Thus, these data can play the greatest value for acquiring a satisfactory need. AI models, including artificial neural network (ANN) and different machine learning (ML) methods, including deep learning (DL) [182], are data-driven intelligence, which has been applied for AI-based condition-based maintenance. M. S. El-Abbasy developed five models based on the ANN technique for the prediction of the oil and gas pipelines condition by in-line inspection dataset from major oil and gas companies in Qatar [183]. It was found that the ANN technique provides better results compared with regression models previously developed. Moreover, they concluded that the corrosion growth rate increased with the increase of metal loss and decreased with the careful maintenance of pipeline cathodic protection. A hybrid intelligent model named PCA-CPSO-SVR, which combines PCA, SVR and chaos particle swarm optimization (CPSO), is proposed for corrosion rate improvement of pipelines [184]. PCA plays a role in reducing data dimension and screening out the main variables of corrosion influencing factors, such as temperature, liquid holdup, etc. The CPSO algorithm was used to enhance the accuracy of the algorithm in support vector regression. From the investigation, higher prediction accuracy was obtained by this hybrid method, whereas it takes the consumption of running time. In addition, an advanced intelligence framework has recently been developed by Seghier et al. [185] for predicting maximum pitting corrosion depth in oil and gas pipelines. Six models containing ANN, multivariate adaptive regression splines, M5 tree, kriging, locally weighted polynomials, and extreme learning machines were completely applied in this framework. The maximum pitting corrosion depth of pipelines located in different environments was sent to the AI models in terms of training and testing. The relationship between the maximum pitting depths and other probable factors inducing the pitting growth process such as the pipeline age, as well as the characteristics of the surrounding environment were carried out. Moreover, ML is a recent emerging-computation intensive analysis method to determine the failure risk of oil and gas pipelines. Pipelines were divided into different failure risk levels according to the failure probability. Eight machine learning algorithms like the random forest, LightBoost (LGBoost) and eXtreme Gradient Boosting (XGBoost), etc., were used and their performance is evaluated by confusion matrix in [186]. It is revealed that machine learning algorithm performs a greater computational efficiency than the physics-based approach, whether deterministic and probabilistic. Moreover, a time-dependent corrosion defect depth growth model of the corroded pipeline based on machine learning was established by using historical operation parameters [187]. A feedforward subspace clustering neural network (SSCN) and particle swarm optimization (PSO) were linked with the model. Furthermore, (CNN) is a kind of feedforward neural network with convolution computation and depth structure. It is one of the representative algorithms of deep learning. It imitates the visual perception mechanism of humans and suitable to work with large volumes of input data. CNN has high efficiency in practice and is guaranteed when working with big data. A work conducted by [188] identifies defects and further condition assessment in the pipeline with CNN. A small database to determine whether there are defects in the pipeline has also been proven to be effective. Another kind of deep learning is reinforcement learning (RL). A model-free RL algorithm based on Q learning was proposed in [189] for condition-based maintenance management of a dry gas pipeline. In this process, a physical-based corrosion degradation model was established as the environment to interact with a decision-maker agent, which obtains information periodically from the pipeline through inspection and determines maintenance action and executes that action, subsequently. After training and testing, the results of the proposed algorithm indicate an improvement in the reliability and showed a 58% reduction in maintenance costs compared to the periodic maintenance regime. However, the limitations of a current agent, such as finite and discreet state and action spaces or the trustable deep RL solution, are further content to investigate. Apart from this, due to the large amount of data and complex environments, combing model-driven and data-driven methods for CBM is necessary. To study the threat of stray current corrosion defect on pipelines, the investigation was conducted by Liu et al. [190]. They used parametric analysis results as database calculated by finite element software to develop a three-layer feedforward artificial neural network for failure pressure prediction. Through experimental burst test results and results of previous failure pressure estimation models verfied by each other, the failure pressure of high-strength pipes with stray current corrosion defect was predicted. It remains challenging to have an adaptive self-evaluated AI model/platform for condition assessment and condition-based maintenance based on pipeline historic, in-situ and environmental data.

### 4.3. Integrated Data Management System and Cloud-Based Management

Despite the emergence of various advanced technologies focusing on the design of intelligent PIG tools, there is no other data analysis or mining method to assess the occurrence of cracks and particle deposition. This is except for the pigging technologies occurring in pipelines [191]. Pigging is usually carried out in time to obtain pipeline infrastructure information, whereas it would result in time loss and maintenance costs. Hence, it is vital to decide when and where to implement the inspection process since this pigging operation is performed periodically. Moreover, it is impossible to identify the leakage or particle deposition as well as the material property data at the initial stage because the existing pigging process cannot predict it. In addition, it will lead to greater risks during oil transportation if cracks, leakages or blockages begin to form inside the oil pipeline after performing the pigging process [192]. This would not be noticed until the next planning for that particular pipeline section arriving.

Although inspection techniques are usually better at locating leaks, they cannot be used continuously. On-line monitoring technology is a better supplementary method. Sensors inserted or connected to the pipe are connected to the software system so that pipe parameters can be continuously monitored and analyzed. Thus, a more reliable, precise, robust, and effective system for inspecting leaks, cracks, or bursts over the pipeline system is required. This system must be supported by all types of data acquired and stored from various field workstations to form a complete database management system. Prediction models and integrity management programs depend on these high-quality analytical data to assist threat identification, such as risk ranking and selection of mitigation techniques. An example is presented in [29]. The procedure has the applicability of material traceability data, including mechanical and chemical properties. Its appearance reflects the merits of an integrated management data system. We can establish a cross-correlation relationship between the early material properties and the test data. Thus, a better analysis for the causes of failure and to predict the probability of defects can be undertaken. The conventional monitoring system incorporates sensors settled on the pipes and communicated through copper or fiber optical cables to central control with limited advantages [193]. Later, RFID (radio of requests) [194,195] and portable robot methods were implemented together with existing wireless applications [196]. The continuous advancement of Internet of Things (IoT) innovations can likewise help achieve completely effective management, where new IoT gadgets will collaborate to more readily handle the status of the appropriation organization [197]. In this case, a critical test is to build a cloud-based network to supervise these IoT devices and cultivate new management institutions to deal with these problems. The data collected by the intelligent terminal is sent to the cloud for data management, analysis, processing, storage, evaluation, prediction and interaction. Based on the big data and cloud network detection and monitoring system in the data management system, it will realize the real-time monitoring of the operation status of the pipe network and carry out the dynamic monitoring and analysis of the big data. Then the treatment of the pipe network emergencies, the analysis of the relevant information and the reasonable planning and design of the pipe network can be undertaken.

As illustrated in Figure 6 from the project report [29], a growing number of web-based technologies such as simulation, defect quantification, case studies and database, and cloud data management are being developed and used for intelligent pipeline inspection and data management.

## 5. Challenges, Problems and Development Trend

Failure models and mechanisms depend on multiple factors. They can be improved through historic and inspection and monitoring data and other information. From the investigation above, various non-destructive evaluation technologies have advantages and limitations for pipeline inspection. More comprehensive modality inspection tools should be developed. For instance, geometric information regarding the length, width, depth, and location of flaw anomalies, which is a critical input for integrity assessment and subsequent effective planning of repair and rehabilitation measures. Nowadays, a variety of special tool configurations are available, and each configuration has been optimized to meet the inspection requirements of the pipeline industry. Multimodality in-line inspection tools can provide important data regarding the flaws and anomalies detected in a pipeline wall. Then the data analysis methods and models are utilized for defect quantification and classification, anticipating the defect growth rate and prediction model for condition-based maintenance. Based on the overview above, the challenges and trends of development are as follows:Multi-physical integration and fusion inspection are expected.

Given the variety of possible defects, it is hard to employ only one NDT technique to achieve high-quality inspection and decision making. Multi-physical field fusion could provide complementary or redundant information from different methods [198]. In other words, multiple information sources in the same or different aspects of the object are fused to reduce uncertainty in order to achieve enhanced detection robustness and accuracy. Compared with using each NDT modality alone, small material faults or defects caused by different failure causes can be detected with higher reliability. For example, conductive materials inspection through ET is useful for near-surface defect detection whereas UT yields volumetric information. Employing both methods and integrating them in the same sensor, the multiple characteristic signals will directly generate from the system to better cover the surface and deep defects [199].
Robotic and instrumental challenge of speed effect and robustness and adaptivity for varied environments

For instance, the inspection speed of MFL is limited to 5 m/s since motion-induced eddy currents in the high-speed inspection [33]. Moreover, the inspection speed of UT and EMAT methods is also limited to 2.5 m/s because the propagation velocity of ultrasonic waves (5.9 km/s in steel pipe) is lower than that of electromagnetic waves. For ECT, a lower excitation frequency requires a longer detection time, which reduces the detection speed. Therefore, this method can only detect ID defects due to the frequency-dependent skin depth effect in high-speed detection. However, the passive detector moves by the fluid in the pipe, the velocity is not uniform, and there will be a mutation. Moreover, other variables such as attachment and lift off will affect the characteristics of electromagnetic signals as well. Under these circumstances, how to improve the inspection robustness and adaptivity for varied environments is a big challenge for active pipeline robot and soft robot for ILI.
Accuracy of location and sizing of defect detection, classification, and quantification

The instrumental systems are regularly found to be stuck during the operation because of the complex shape of the pipeline, bends or magnetization force in MFL PIG systems. Thus, it causes data discontinuity and data analysis will lead to inaccurate positioning [200]. In recent years, an emerging technology is to add an auxiliary positioner to smart devices for accurate locating. These consist of odometer wheel, inertial measurement unit (IMU), etc. However, these auxiliary apparatuses are time-dependent, and the cumulative error will increase with time-consuming. In addition, as described in Section 5, no tools can achieve 100% signal diversity detection probability in different environments. Therefore, the accuracy of location and sizing of defect detection [201], classification, and quantification would be a research spot in the future instrumental system.
Multiple parameter measurement and characterization, e.g., integration of inspection and structural health monitoring, e.g., defect detection and stress characterization

For some accident-prone areas [30] or suspicious areas detected by the inspection tools, multi-parameter measurement and real-time monitoring are very necessary [202]. Moreover, due to the promotion of the internet of things and physical network systems, RFID tag antennas and sensors have been widely used in the field of antennas and sensors. These type of sensors have potential applications in SHM [203] because of their passivity, wireless, simplicity, compactness and multimodality, especially in the life cycle of large infrastructure [204]. Thus, it is very suitable for pipeline monitoring. Meanwhile, the applications of characterizing materials in terms of microstructure, condition, and properties are important tasks. The birth of the micromagnetic multiparametric microstructure and stress analyzer (3MA) and its broad range of applications indicate that it is a possible method to quantitatively determine the mechanical properties of materials [205]. However, applications are primarily available in the laboratory and some specialized systems can be used for semi-finished products, e.g., strip steel, heavy plates, or ferromagnetic steels [206]. Moreover, the physical mechanism of different operation configurations, the relationship between the excitation mode and the measured multi-feature quantity are still needed to be further studied. It is the inevitable trend of future development to integrate inspection and structural health monitoring for defect detection and microstructure characterization. Meanwhile, the related physical principles for better understanding characterization relationship and sensor/system design also have broad research prospects and markets. For example, flexible sensors [113] can be applied for inspection and monitoring [207].
Lifetime prediction, AI-assisted condition-based maintenance through intelligent data management and security

Defects growth calculation and lifetime prediction can be applied in future inspection planning, excavation costing planning and a pipeline operator’s decisions on maintenance or replacement. This information can be obtained by appropriate mathematical physics models or artificial intelligence technologies with intelligent data management. According to the datasets, pipeline safety planning and management should take place without delay if the defect growth rates are high or the predicted lifetime is coming to an end [173].

Despite the convenience of information sharing, security [208] and accessibility of data management should not be ignored. If necessary data are missed or changed at will, this will cause data failure and bring serious consequences. The stability and response speed of massive data transmission [203] will also face great challenges. Unstable data transmission will lead to partial data loss, thus affecting the integrity of the data. At the same time, the slow transmission speed would waste time and reduce efficiency as well. More research on remaining life prediction methods, models, and theories in associated with AI and big data will be required.

## 6. Conclusions

In this paper, four main procedures of a pipeline integrity management technology have been discussed. The significance of pipeline inspection is firstly highlighted under the discussion of different defect information and types of failure. In addition, the basics of non-destructive evaluation technologies and the use of PIGs or other robots as the application are reviewed in detail. Furthermore, data analysis methods and models are utilized for defect quantification and classification, and the prediction of failure and maintenance are reviewed. Then the challenges, problems, and development trends of the instrumental system have been discussed and analyzed. In conclusion, the main objectives of pipeline integrity management should first obtain accurate information through detection. Then all reasonable anticipated defects and safety threats during pipeline design, construction and operation should be identified and evaluated. Afterwards, appropriate measurements need to be taken to prevent failures that could cause damage to pipelines. Effective techniques require to predict and estimate the future condition or lifetime of pipelines. Thus, condition-based maintenance and intelligent decision making can be determined. According to this, further research on pipeline inspection must continue to evolve. The research directions should not only focus on innovation of detection theory but also improve integration of multi-function detection systems and data management. This paper has undertaken a systematic review from inspection, robot-based data acquisition, data management and decision making.

In the future, multi-physical field fusion inspection is likely to be a hot spot for sensor research and development in detection tools. Based on this, new materials, new sensor structure design, and higher requirements are needed. At the same time, the challenge of speed effect and robustness and adaptivity for varied environments need to be solved urgently. In addition, more reliable and efficient signal processing and data analysis approaches need to be developed to remove noise and accurately evaluate defects. Accuracy of location and sizing of defect detection, classification, and quantification will require higher demand with the equipment updating. In addition, when the inspection and structural health monitoring are integrated, multi parameters are required to characterize the defects and stress. Then, the macro signal and micro mechanism are organically combined, which makes the non-destructive evaluation more scientific. After that, intelligent life prediction and auxiliary maintenance are indispensable. Last but not least, based on big data and cloud network management systems, privacy security and accessibility cannot be ignored as well.

## Figures and Tables

**Figure 1 sensors-21-03862-f001:**
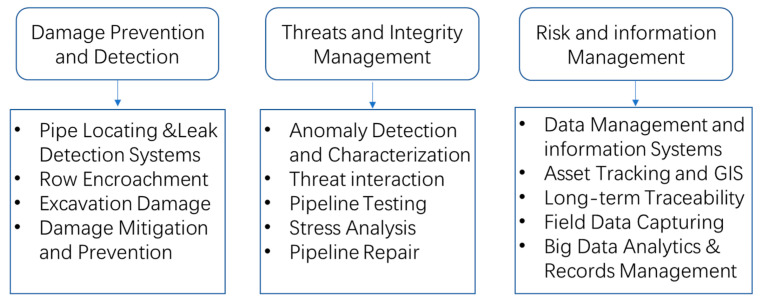
Technologies categories and research areas. Adapted with permission from ref. [29]. 2021 Copyright Khalid Farrag.

**Figure 2 sensors-21-03862-f002:**
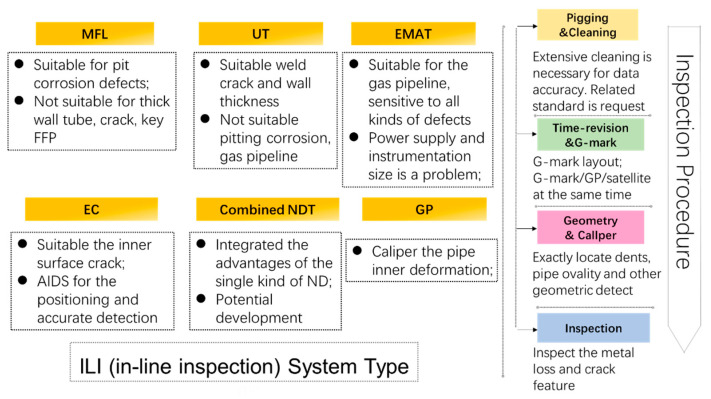
Structure schematic of internal inspection (ILI) system. Reprinted from ref. [110].

**Figure 3 sensors-21-03862-f003:**
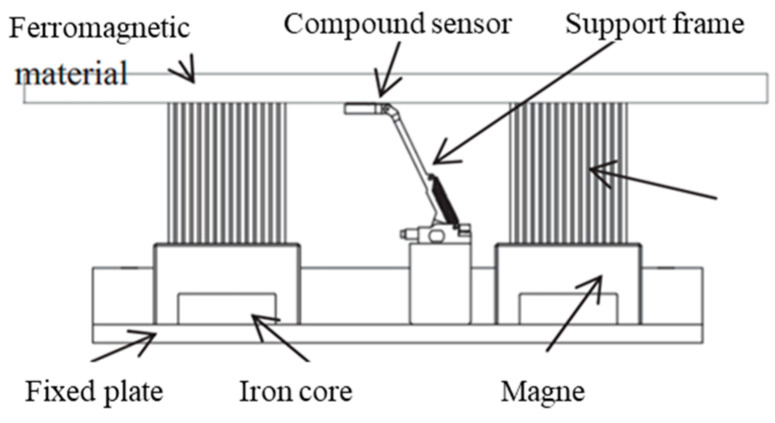
Magnetoelectric compound sensor detection device. (Reprinted with permission from ref. [124]. Copyright 2019 Xiaoting, G., et al).

**Figure 4 sensors-21-03862-f004:**
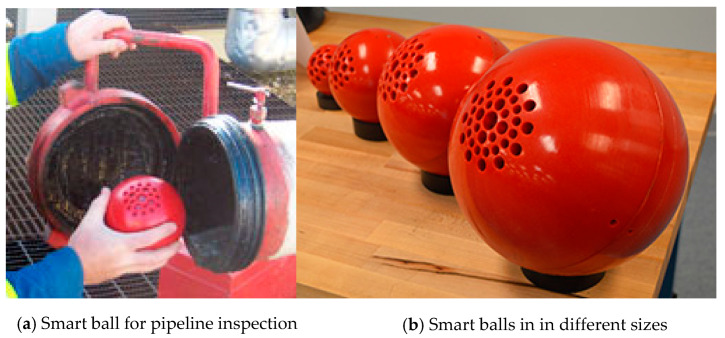
Pure Technologies, Ltd. (Calgary, Alberta) Smart-Ball. (Reproduced with permission from ref. [130]. Copyright 2021 American Society of Mechanical Engineers ASME).

**Figure 5 sensors-21-03862-f005:**
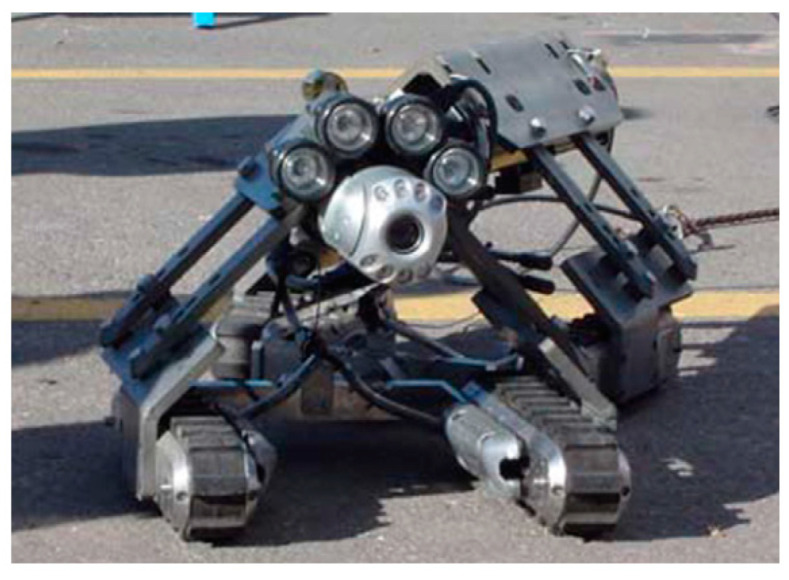
Closed-circuit television camera (CCTV) pipeline inspection from [144]. (Reprinted with permission from ref. [144]. Copyright 2021 Elsevier).

**Figure 6 sensors-21-03862-f006:**
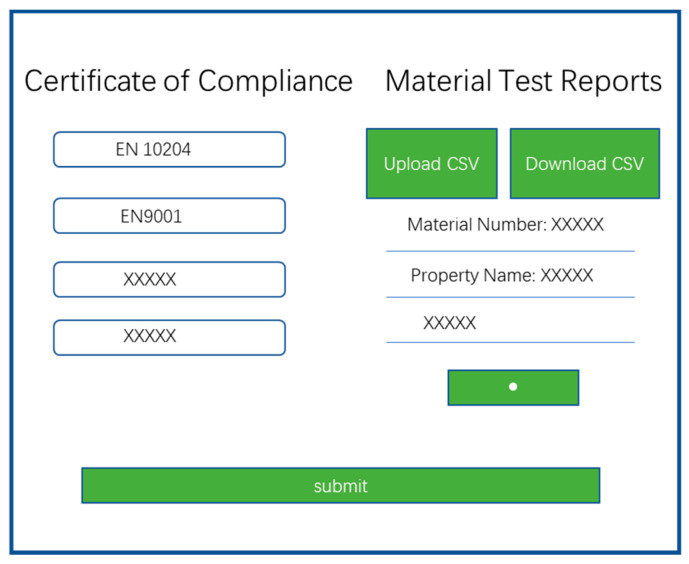
Material traceability record. (Adapted with permission from ref. [29]. 2021 Copyright Khalid Farrag).

**Table 1 sensors-21-03862-t001:** Defects of steel pipelines produced during prefabrication and their causes [4,5,7,8,9,26,27].

Defects	Location	The Reason for the Formation
shrinkage cavity	near-surface	the last solidified of molten metal shrinks
casting hot crack	internal and external surfaces	stress due to different solidification rate
stoma	surface or near-surface	the gas is retained when the metal solidifies
inclusion	surface or near-surface	impurities were added into the casting process
cracks	surface	the surface depression is discontinuous and elongated during rolling
layered	surface or near-surface	inherent defects are elongated and flattened during the rolling
fold	surface	excess material covering and pressing into surfaces
heat treatment crack	surface	uneven heating or cooling
coating crack	surface	residual stress release

**Table 2 sensors-21-03862-t002:** Defects of pipelines produced during operation and their causes [4,5,15,16,19,28].

Defects	Location	The Reason for the Formation
fatigue crack	surface	periodic stress application below the ultimate tensile strength of the material
stress corrosion cracking	surface or near-surface	the combined action of tensile static load and corrosive medium
hydrogen-induced cracking	surface	tensile or residual stress interacts with the hydrogen-rich medium
corrosion	surface	interaction of corrosive medium and alternating stress

**Table 3 sensors-21-03862-t003:** Comparison of various technologies [3,4,112,113].

Inspection Strategy	Merits	Limitations
MFL	without the need for pre-processing, easy online detection, highly automated for detecting various types of defects	relative movement between MFL probes can distort the profile of MFL signals, not good in poorly magnetized materials like stainless steel
EC	sensitive to multiple parameters; wider operating temperature range, suitable for small diameter pipelines inspection due to smaller sizes for probes, lightweight and convenient to be located on micro-robots, and more economical	the depth of penetration is dependent on the frequency of the AC current applied to the coil, suffers from the lift-off effect
UT	high penetration depth and suitable for testing all kinds of materials and their properties, thickness and external corrosion can be estimated	easily affected by dense highly attenuating muds and casing scales, not sensitive enough to small features
ECPT	high spatial resolution, fast detection response, and wide range detection, intuitive and reliable	affected by the surface emissivity, the infrared camera blocks the view, the internal crack detection is limited
MBN	high sensitivity to microstructure and stress state of materials, fast detection, and harmless to the operator	difficult to find a consistent behavior of the MBN signal, can only be pick up near the surface of the materials
RT	permanent images record, require no surface treatment or insulation removal, and less sensitive to external deposits	potential harm to the human body and cause environmental pollution
AE	applicable to dynamic detection and large region can be tested	cannot provide the condition of the static defect and it is a contact measurement method
PT	sensitive to opening surface cracks and not affected by workpiece geometry and defect direction	penetrant process is complex and requires cleaning operation. It can cause environmental pollution as well
MT	high detection sensitivity and it can intuitively display the position, shape, size, and severity of the defect	the procedure is complicated and only for surface and near-surface defects of ferromagnetic materials
VT	economical and easy to operate	The test results are easily affected by human factors and only for surface discontinuities

**Table 4 sensors-21-03862-t004:** Emerging commercial Pipeline Inspection Gauge (PIG) technologies.

PIG Type	Technical Function	Image
GP	To collect information relating to the physical shape or geometry of pipelines	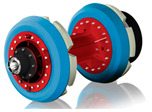 ROGEO Untouched GP. Reprinted from ref. [110].
MFL	Suitable for the pipe diameter range of 76–1422 mm and integrated for super high resolution to identify and size significant corrosion	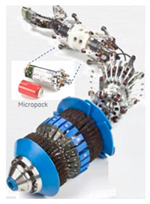 GE PII MagneScan SHR MFL [4] (Reproduced with permission Elsevier)
UT	Special configuration unites metal loss and cracks detection, available for pipeline size 20″ and above	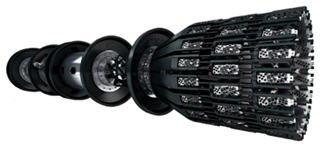 NDT-GLOBAL LineExplorer UCM. Reprinted from ref. [110]
EMAT	High reliability inspection and accurate continuous measurement of critical crack anomalies, coating disbandment	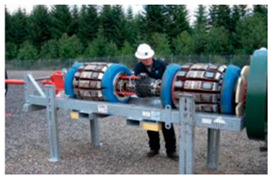 ROSEN RoDD EMAT [119,126]. Reprinted from ref. [119]
EC	Integrated with deflection sensors that enable for simultaneous measurement of internal pipe profile and metal loss	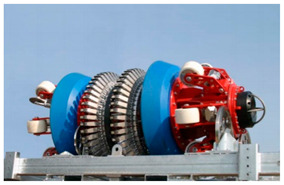 ROSEN EC [110,127]. Reprinted from ref. [110]
	Has electromagnetic sensors embedded into the polyurethane. The array of electromagnetic sensors detects shallow internal corrosion and fatigue cracking (SICC) in dry gas or multiphase pipelines	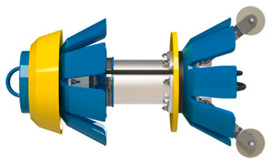 I2I eddy current Pioneer (Reprinted with permission from ref. [120]. Copyright 2021 I2I Pipelines.)
	MWM-Array technology is used for high-resolution imaging of internal corrosion, internal initiated and relative stresses can be provided	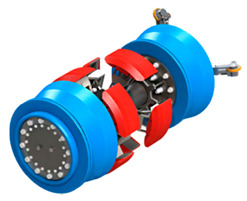 JENTEK ILI Tool [121] (Reproduced with permission ASME Press)
Integrated Function	enable multiple data acquisitions for pipeline integrity with a single run, reduces inspection costs and workload	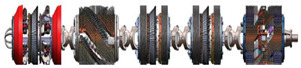 TDW (DEF+SMFL+MFL+LFM+EMAT). Reprinted from ref. [110]
Specific Function	Cathode protection current measurement ILI system which can capture data that verifies the effectiveness of Cathode Protection	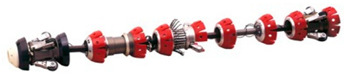 Baker Hughes CPCM ^™^^.^ Reprinted from ref. [110]
MEC	Inspection of compound pipelines with stainless steel and carbon steel in two layers	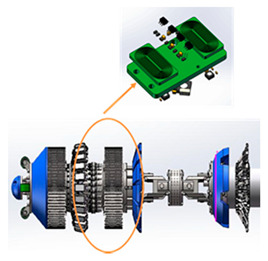 Shenyang Academy of Instrumentation Science MEC Tool

**Table 5 sensors-21-03862-t005:** Advice for the choice of commercial PIG. (Reprinted from ref. [110]).

Consideration Parameters	Metal Loss Features	Crack Features	Deformation and Geometry
Gas/Liquid medium Operation pressure, High-flow velocity, Wall thickness, Pipe grade, Internal coat, Multi/Dual-diameter Cathode Protection (CP) system, Ambient	General corrosion, Pitting, Pinholes, Axial groove, Lamination, Wall thinning, Narrow axial external corrosion	Hook/seam weld crack, Hydrogen induced crack, Circumferential crack, Fatigue crack, Shrinkage crack, Lack of fusion, Crack in dents, Stress corrosion cracking (SCC)	Plain dent, Dents with metal loss, Small dents, ID expansions, Buckle/wrinkle, Bend, Bending strain Centerline mapping
Advice of choice	MFL	UT/EMAT	GP/EC

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
