# Peer review of "Pipeline In-Line Inspection Method, Instrumentation and Data Management"

_sensors, 2021, doi:10.3390/s21113862_

Round 1

Reviewer 1 Report

This is a well-written paper presenting a review of methods, instrumentations, and data management strategies that are used in pipeline in-line inspection. The references to previous studies regarding this issue are highly valued. Even though this review can be considered as a good working document (technical report) as it summarizes to date developed experimental techniques, it is unclear to me what purpose the paper should serve. The content of this review can be found in some Ph.D. dissertation literature reviews or scientific textbooks on pipeline inspection. For these reasons, I do not think that the paper can be published in its present form. Without any further motivation, briefly repeating already published works cannot justify the publication of a paper.

Author Response

All the reviewers' comments have been addressed.

Reviewer 2 Report

The manuscript is a review of the state-of-the-art in pipeline in-line inspection methods and the connected themes. The author provides an extensive bibliography and summarize in a somewhat effective way the cited works. The manuscript does not contain original considerations, but the review analysis is well within the scope of the journal. The language must be double-checked since the list of corrections is significant, albeit acceptable considering the length of the paper. The overall impression is positive, and it is the opinion of the reviewer that this work should be published after some minor revisions are performed.

A list of the found language errors follows:

l.26  firstly -> first
l.27  remove "detail"
l.39  that might be occurred -> might occurred
l.46  appear -> contain
l.55  are commonly occured -> commonly occurr
l.60  remove "that is"
l.81  form several types including -> include
l.84  extend -> extent
l.105 remove "as the following aspects"
l.110 Section 4 reviews the -> In section 4 the
l.110 utilized -> reviewed
l.150 and, data -> and data
l.170 diploe -> dipole
l.189 diffraction and the -> diffraction, and
l.211 remove "GWUT for monitoring and off-line inspection?"
l.246 It affected -> It is affected
l.248 remove "and then"
l.312 They caused -> They are caused
l.342 is produced -> produces
l.361 remove "and"
l.368 remove "while"
l.389 figure too large, rescale
l.430 appropriate -> properly
l.485 which not -> which is not
l.535 the requirements of the construction -> construction requirements
l.579 "(GAN)" repeated
l.604 investigates -> investigation
l.621 are served to account -> account
l.625 are represented -> represent
l.650 distribution, which is -> distributions, which are
l.721 to identify -> identifies
l.741 challenges -> challenging
l.756 remove "scheduling"
l.795 "After ... making": missing verb.
l.807 remove "there are"
l.809 remove "specially"
l.845 "Through ... positioning": unclear, rephrase.
l.849 as time-consuming -> with time-consuming
l.870 can used -> can be used
l.873 remove "are"
l.899 firstly -> first
l.902 in detail reviewed -> reviewed in details

Author Response

The comments have been addressed.
